

# Estimating Marine Carbon Uptake in the Northeast Pacific Using a Neural Network Approach

Patrick J. Duke[1], Roberta C. Hamme[1], Debby Ianson[2,1], Peter Landschützer[3], Mohamed M. M. Ahmed[1,4], Neil C. Swart[5,1], Paul A. Covert[2]

[1]School of Earth and Ocean Sciences, University of Victoria, Victoria, BC, Canada
[2]Institute of Ocean Sciences, Fisheries and Oceans Canada, Sidney, BC, Canada
[3]Flanders Marine Institute (VLIZ), Ostend, Belgium
[4]Education and Research Group, Esri Canada, Calgary, AB, Canada
[5]Canadian Centre for Climate Modelling and Analysis, Environment and Climate Change Canada, Victoria, BC, Canada

*Correspondence to*: P. J. Duke (pjduke@ucalgary.ca)

**Abstract.** The global ocean takes up nearly a quarter of anthropogenic $CO_2$ emissions annually, but the variability of this uptake at regional scales remains poorly understood. Here we use a neural network approach to interpolate sparse observations, creating a monthly gridded seawater partial pressure of $CO_2$ ($p$$CO_2$) data product from January 1998 to December 2019, at 1/12°x1/12° spatial resolution, in the Northeast Pacific open ocean. The data product (ANN-NEP; NCEI OCADS Record ID: BGSH2HNRP) was created from $p$$CO_2$ observations within the 2021 version of the Surface Ocean $CO_2$ Atlas (SOCAT), and a range of predictor variables acting as proxies for processes affecting $p$$CO_2$ to create non-linear relationships to interpolate observations at a spatial resolution four times greater than leading global products and with better overall performance. In moving to a higher resolution, we show that the internal division of training data is the most important parameter for reducing overfitting. Using our $p$$CO_2$ product, wind speed, and atmospheric $CO_2$, we evaluate air-sea $CO_2$ flux variability. On sub-decadal to decadal timescales, we find that the upwelling strength of the subpolar Alaskan Gyre, driven by large-scale atmospheric forcing, acts as the primary control on air-sea $CO_2$ flux variability ($r^2 = 0.93$, $p$ <0.01). In the northern part of our study region, divergence with atmospheric $CO_2$ is enhanced by increased local wind stress curl, enhancing upwelling and entrainment of naturally $CO_2$-rich subsurface waters, leading to decade-long intervals of strong winter outgassing. During recent Pacific marine heatwaves from 2013 on, we find enhanced atmospheric $CO_2$ uptake (by as much as 45%) due to limited wintertime entrainment. Our product estimates long-term surface ocean $p$$CO_2$ increase at a rate below the atmospheric trend (1.4±0.1 µatm yr$^{-1}$) with the slowest increase in the center of the subpolar gyre where there is strong interaction with subsurface waters. This mismatch suggests the Northeast Pacific Ocean sink for atmospheric $CO_2$ may be increasing.

## 1 Introduction

As countries around the world consider updating their carbon emission reduction commitments (United Nations Environment Programme, 2022), we require a better understanding of global carbon sinks and how they may be shifting under climate change. The global ocean takes up nearly a quarter of anthropogenic carbon dioxide ($CO_2$) emissions annually



(Friedlingstein et al., 2022) but the temporal and spatial variability of the marine sink remains unclear on decadal or longer timescales (McKinley et al., 2011; Fay and McKinley, 2013; Wanninkhof et al., 2013; Gruber et al., 2023). Potential future changes in the marine sink associated with climate change are also unclear (O'Neill et al., 2018). Extending the spatial and temporal coverage of partial pressure of $CO_2$ in seawater ($pCO_2$) observations can help address this knowledge gap (Aricò et al., 2021). Benefitting from the increasing abundance of $CO_2$ measurements at sea and community synthesis efforts (e.g., through the Surface Ocean $CO_2$ Atlas (SOCAT)), a variety of interpolation approaches have evolved capable of creating continuous observation-based estimates of $pCO_2$ (Denvil-Sommer et al., 2019; Zhong et al., 2022; Laruelle et al., 2017; Nakaoka et al., 2013; Chen et al., 2019; Ritter et al., 2017; Landschützer et al., 2013). However, their global focus and coarse resolution limits their interpretation at regional scales (Olivier et al., 2022). Only recently, higher resolution regional $pCO_2$ maps have been developed for the California current system (Sharp et al., 2022) to overcome the limitations of coarse global scale $pCO_2$ products. These seawater $pCO_2$ products, combined with wind speed and atmospheric $pCO_2$, have informed regional to global air-sea $CO_2$ flux estimates of multiyear variability (Landschützer et al., 2019, 2016, 2015; Wang et al., 2021; Hauck et al., 2020).

No high-resolution observation-based air-sea $CO_2$ flux estimate currently exists for the North Pacific Ocean. The Northeast Pacific Ocean has been characterized as a net annual sink for atmospheric $CO_2$ (Wong et al., 2010, p.201; Franco et al., 2021; Duke et al., 2023; Sutton et al., 2017). The region is divided by two dominant oceanographic features, the Alaskan Gyre system to the north, and the North Pacific Current to the south (Franco et al., 2021). With respect to surface ocean carbon measurements, the Alaskan Gyre system remains extremely sparsely sampled. The seasonal air-sea $CO_2$ flux of the gyre has been described as being strongly influenced by gyre upwelling with outgassing in the winter and uptake in the summer (Brady et al., 2019; Palevsky et al., 2013; Chierici et al., 2006). Along the easternmost part of the North Pacific Current, most of our understanding comes from a limited region; the Ocean Station Papa mooring at 50°N, 145°W (Sutton et al., 2017), and the Line P program (Freeland, 2007). This region has well documented seasonal cycles (Sutton et al., 2017), interannual variability (Wong and Chan, 1991; Wong et al., 2010), and long-term trends (Franco et al., 2021; Sutton et al., 2019). $CO_2$ uptake is mainly driven by direct ventilation of the shallow upper water column, with a small seasonal change in surface ocean $pCO_2$ (Wong et al., 2010; Sutton et al., 2017). The estimated long-term trend in surface ocean $pCO_2$ appears to be increasing at less than the atmospheric rate (Franco et al., 2021).

Understanding what drives air-sea $CO_2$ fluxes on seasonal, interannual, and decadal timescales in the Northeast Pacific Ocean will inform how the regional sink may change in future. This region is already experiencing persistent marine heatwaves with dramatic temperature anomalies observed during 2014 to 2016 and 2018 to 2020 events (Freeland and Ross, 2019; Bond et al., 2015), with future events predicted to become longer-lasting, more frequent, more extensive, and more intense (Frölicher et al., 2018). The impact of large-scale climate-driven decadal oscillations on the marine carbon system is just beginning to be explored in models (Hauri et al., 2021). Furthermore, this region has been targeted as a potential site of marine carbon dioxide removal, as a negative emissions technology aimed at meeting emission reduction goals continues to grow in interest and investment (Cooley et al., 2022). Some proposed approaches look to artificially stimulate biological



carbon drawdown (GESAMP, 2019; NASEM, 2021). The Northeast Pacific Ocean, as an iron-limited high-nutrient low-chlorophyll region (Dugdale and Wilkerson, 1991; Aumont et al., 2003; Martin et al., 1994; Freeland et al., 1984), has already been the location of geoengineered biological carbon drawdown experiments (Boyd et al., 2007, 2005; Wong and

Johnson, 2002; Ianson et al., 2012). Thus, a firm understanding of processes driving carbon fluxes and the establishment of environmental baselines in the region is critical.

Our aim is to investigate drivers of air-sea $CO_2$ flux variability in the Northeast Pacific (NEP) Ocean, building a novel regional high-resolution artificial neural network (ANN) approach adopted from an existing global setup (Landschützer et al., 2013). In Section 2, we describe the creation of a gridded $p$CO$_2$ data product (herein referred to as ANN-NEP) monthly

from January 1998 to December 2019 at 1/12°x1/12° spatial resolution in the Northeast Pacific open ocean (approximately 9 km by 5km; latitude by longitude). In Section 3, we show that the high-resolution regional $p$CO$_2$ product is robust enough to recreate training observation data while generalizing well compared to independent withheld observation data. We also show that stepping to a higher resolution regionally with appropriate tuning of the internal training and evaluation data ratio does not hinder product performance. In Section 4, our results show that the upwelling strength of the subpolar Alaskan Gyre and

surface ocean connectivity to subsurface waters act as the primary control on air-sea $CO_2$ flux variability in our study area. We conclude by calculating long-term trends in surface ocean $p$CO$_2$ and carbon uptake, examining trends relative to connectivity to subsurface waters.

## 2 Data and methods

Our study area comprises the region between latitudes 45°N and 62°N and longitudes 120°W and 155°W (Figure 1), with the

open-oceanic/coastal boundary defined as 300 km offshore following Laruelle et al. (2017). This work represents a four times increase in spatial resolution over previous multiyear global open ocean products, usually coarser than 1/4 of a degree (Landschützer et al., 2020b). The increased resolution derives from high-resolution predictor data used to create the product (Table 1). To interpolate the existing $CO_2$ observations in this domain, we adapt the artificial neural network (ANN) self-organizing-map-feed-forward-network (SOM-FFN) approach developed by Landschützer et al. (2013, 2014). In a first step,

the method divides the region of interest into dynamic zones with similar biogeochemical features (i.e., SOM biogeochemical provinces), using a self-organizing map approach. In a second step, a feed-forward neural network is used for interpolating $p$CO$_2$ observations in each of the pre-determined provinces of step one. Specifically, non-linear functional relationships are created between $p$CO$_2$ observations (or neural network target data), where they exist in our study domain, and independent predictor variables (or neural network input data) that are known to drive the marine carbon cycle (see

Section 2.1 below). Once the relationships are established, they can be applied where no observations exist to fill space/time gaps and create continuous sea surface $p$CO$_2$ maps from 1998-2019.



## 2.1 Predictor data

The chosen predictor variables for this study (Table 1) had all been used previously in observation-based $p$CO$_2$ interpolated products (Denvil-Sommer et al., 2019; Zhong et al., 2022; Landschützer et al., 2014; Gregor et al., 2018; Telszewski et al.,

2009). Sea surface temperature (SST) comes from the satellite-based European Space Agency Climate Change Initiative (Merchant et al., 2019; ESA Sea Surface Temperature Climate Change Initiative (SST_cci): Level 4 Analysis Climate Data Record, version 2.1), as well as Chlorophyll-*a* concentration which served as a proxy for biological processes (ESA Ocean Colour Climate Change Initiative (Ocean_Colour_cci): Global chlorophyll-a data products gridded on a geographic projection, Version 5.0). Remaining physical process predictor data (e.g., sea surface salinity (SSS), sea surface height

(SSH), and mixed layer depth (MLD)) are obtained from Copernicus Marine Environment Monitoring Service global ocean eddy-resolving reanalysis (Global Ocean Physical Reanalysis Product, E.U. Copernicus Marine Service Information GLOBAL_REANALYSIS_PHY_001_030). Jointly assimilated observations include satellite altimeter data and *in situ* vertical profiles of temperature and salinity informing the MLD reanalysis product (Table 1). The ocean general circulation model is based on the Nucleus for European Modelling of the Ocean (NEMO) platform, driven at the surface by the

European Centre for Medium-Range Weather Forecasts ERA-Interim winds (Jean-Michel et al., 2021). Both Chlorophyll-*a* and mixed layer depth were log10-transformed to produce a distribution of values closer to normal before being used in either SOM-FFN step. Atmospheric mole fraction of CO$_2$ ($\chi$CO$_2$) is derived from data produced by National Oceanic and Atmospheric Administration Earth System Research Global Monitoring Laboratory (https://gml.noaa.gov/ccgg/globalview/). Finally, the monthly $p$CO$_2$ climatology of Landschützer et al. (2020) was used as an additional input parameter solely for

defining the SOM biogeochemical provinces.

## 2.2 $p$CO$_2$ observations

ANN target $p$CO$_2$ data come from the Surface Ocean CO$_2$ Atlas (SOCAT) v2021 (Bakker et al., 2016), as well as additional data from the Fisheries and Oceans Canada February 2019 Line P cruise (https://www.waterproperties.ca/linep/; Figure 1c). Sea surface CO$_2$ fugacity ($f$CO$_2$) was converted to sea surface $p$CO$_2$ (Körtzinger, 1999). $p$CO$_2$ observations were bin-

averaged into monthly, 1/12° latitude by 1/12° longitude grid cells computing the mean and standard deviation within each grid cell. Of the 8,712,264 grid cells that represent the surface ocean gridded in three dimensions over 264 months (1998–2019) at 1/12°x1/12° resolution in the study area, just 0.39% have an associated gridded $p$CO$_2$ value (Figure 1).

## 2.3 Evaluation

In constructing the optimal ANN architecture, a series of SOM-FFN tuning tests were conducted comparing ANN output to

training and independent withheld data. ANN performance for each tuning test was evaluated using five statistical metrics: root mean squared error (RMSE), coefficient of determination ($r^2$), mean absolute error (MAE), mean bias (calculated as the mean residual), and the slope of the linear regression ($c_1$) between the ANN and the corresponding gridded SOCAT $p$CO$_2$





observations. Independent withheld data came from randomly selected SOCAT data using associated expocodes corresponding to unique complete underway cruise tracks or mooring deployments. We tested 100 random independent

withheld data splits and selected one representative of basin-wide observational coverage (summer/southern sampling bias), with winter, spring, and fall data present (Figure 1; supplementary Figure 1). These independent withheld data represented approximately 5% of the total study area gridded $p$CO$_2$ data, with coverage during all seasons over a range of latitudes (supplementary Figure 1).

### 2.4 Neural network construction

SOM-FFN tuning tests occurred in series using the MATLAB Neural Network Toolbox, with sequential improvements impacting future tests. Optimization of the SOM-derived biogeochemical provinces involved trial-and error testing of various parameters including SOM biogeochemical province count, predictor variables choice, and static or varying province shape with each timestep (Landschützer et al., 2013). The choice of four SOM biogeochemical provinces represented the lowest number of SOM biogeochemical provinces for a typical clustering structure to emerge, while keeping the ratio of

gridded $p$CO$_2$ observation to the total grid cells within each province similar (0.38±0.06%). The best SOM predictor variables were SST, SSS, MLD (Table 1), and the Landschützer et al. (2020a) $p$CO$_2$ climatology. We did not normalize predictor data (e.g., force a mean of 0 and standard deviation of 1), implicitly weighting SOM predictors toward the $p$CO$_2$ climatology as was done in Landschützer et al. (2013). As a result, our dynamic provinces follow the seasonal variations in the $p$CO$_2$ climatology (Landschützer et al., 2020a). Thus, non-static provinces, that changed shape from one month to the

next over a climatology, proved the most useful in clustering seasonal cycle variability. This clustering does lead to clearly unphysical fronts as an artifact of the approach.

In reaching an optimal FFN architecture (i.e., number of inputs, number of hidden layers and neurons in each hidden layer), trial-and error testing of tuning parameters explored predictor variable choice, FFN training algorithm and activation functions, pre-training to determine the number of neurons in the first hidden layer, introducing a second hidden layer with a

static number of neurons, and changing the internal data division ratio.

To emphasize interannual and longer-term trends within predictor variables (Table 1), we introduced each predictor variable again after deseasonalizing. For each grid cell, the monthly anomaly was calculated by subtracting the climatological monthly mean, removing the seasonal cycle from the data (the same approach is used when looking at anomaly values in our results; Section 4). Where no chlorophyll-*a* satellite data were available, the ANN was run again with the remaining

predictors and output was merged to fill empty grid cells (Landschützer et al., 2014). The Levenberg–Marquardt backpropagation training algorithm and hyperbolic tangent sigmoid activation function (i.e., trainlm and tansig respectfully in MATLAB) were found to deliver the best fit. The number of neurons within the first hidden layer varied by province and the optimal number of neurons was determined in a pre-training run, where we increased the number of neurons parabolically from two up to a number where the ratio between the training sample size to the number of weights did not



exceed 30 (i.e., a number that was determined by trial and error). The best output performance of the pre-training determines
       the best neuron setup which was then further used for the actual ANN training.

       To avoid overfitting, we split all the internal training data into two subsets (i.e., one actual training dataset and one internal
       evaluation dataset). While most studies use a fixed ratio (usually 80:20) between these sets, we used the optimal ratio
       determined by a criterion suggested in Amari et al. (1997) that is dependent on the number of degrees of freedom and hence

varies with the optimal number of neurons determined in the pre-training (see Section 3.4 below). While the training dataset
       is used to reconstruct the non-linear relationship between input data (Table 1) and $p$CO$_2$ observations, the internal evaluation
       data are used to stop the training before the network starts overfitting the training data. Specifically, we stopped the training
       when 6 consecutive iterations did not reduce the network's error compared to internal evaluation data (Hsieh, 2009). The
       addition of a second hidden layer with a static neuron number of five was found to slightly improve performance within the

evaluation metrics.

## 2.5 Cross-evaluation and ensemble

       In order to further decrease the risk of overfitting, we used a 10-fold cross-evaluation approach (Li et al., 2019, 2020) and a
       bootstrapping method (Landschützer et al., 2013). Here, all SOCAT cruises (apart from the independent withheld data;
       Section 2.3) were randomly divided into ten equal subsamples using SOCAT expocodes prior to gridding. One subsample

was used as 10-fold evaluation data (10% of all data), and was excluded from training, while the remaining nine subsamples
       were used together as training data (90% of all data). The cross-evaluation process was repeated ten times, with each of the
       ten subsamples used exactly once as the 10-fold evaluation dataset. We performed ten trainings with each 10-fold training
       data subsample where we randomly split the ANN internal training and evaluation data based on the optimal ratio
       determined through testing (Section 3.4). The robustness and reliability of an ANN has been shown to be significantly

improved by combining several ANNs into an ANN ensemble model (Sharkey, 1999; Linares-Rodriguez et al., 2013;
       Fourrier et al., 2020). The ten different ANN outputs trained on ten different 10-fold training data subsamples were used as
       an ANN ensemble, where the ten outputs were averaged to obtain the final ANN-NEP $p$CO$_2$ product (Fourrier et al., 2020).

## 2.6 Computation of air-sea fluxes

       Using the ANN-NEP $p$CO$_2$ product, the air-sea CO$_2$ flux ($F$CO$_2$), was calculated using Eq. 1:

$F\text{CO}_2 = \propto k\Delta p\text{CO}_2 ,$                                                                    (1)

       based on solubility ($\propto$) as a function of temperature and salinity using the data presented in Table 1 (Weiss, 1974), gas
       transfer velocity ($k$), and the gradient between $p$CO$_2$ in the surface ocean and the atmosphere ($\Delta p$CO$_2$). Here, the gas transfer
       velocity is a function of wind-speed retrieved from Cross-Calibrated Multiplatform ocean surface wind data (Mears et al.,

2019), the temperature dependent Schmidt number specific to CO$_2$, and gas transfer coefficient from Wanninkhof (2014).





Negative (positive) flux values indicate $CO_2$ uptake (outgassing) by the ocean. Uncertainty in the air-sea $CO_2$ flux comes from a 20% uncertainty in k (Wanninkhof, 2014) and the overall product uncertainty in estimated $pCO_2$ ($\theta_{pCO2}$; Eq. 2; see Section 3.2 below). As the uncertainty of $\Delta pCO_2$ is dominated by the uncertainty in estimated surface ocean $pCO_2$, we neglect the small contribution from atmospheric $CO_2$ (<1 µatm; Landschützer et al. 2014).

## 3 Network performance

### 3.1 Evaluation comparing to SOCAT data

Overall, the final high-resolution regional artificial neural network Northeast Pacific $pCO_2$ product (ANN-NEP) obtains good fits with an overall $r^2$ of better than 0.8 and RMSE of around 11 µatm between the estimated $pCO_2$ and the gridded SOCAT $pCO_2$ data across both the training data (Figure 2a), and independent withheld data (Figure 2b). The mean bias is negligible (<0.8 µatm; smaller than observational uncertainty). These results also apply within individual calendar years, and within monthly groupings across all years, indicating that the temporally inhomogeneous data distribution over the time range and between seasons does not have a measurable effect on the estimates (supplementary Table 1). There is no clear spatial structure to the residuals, with no specific region displaying persistently positive or negative residuals (supplementary Figure 2). When compared to local $pCO_2$ mooring data from Ocean Station Papa (which is included in SOCAT; Figure 1; Sutton et al. 2017), the ANN-NEP product also performs well ($r^2 = 0.86$; 133 months; not shown).

The ANN ensemble model approach demonstrated improved performance metrics when compared to each individual ensemble member. Overall, individual ensemble members showed little deviation (RMSE <8 µatm) from the ensemble mean (Figure 2c), with the ensemble mean still improving estimate robustness and reducing overtraining as evident in comparing the final ANN product to independently withheld data (Figure 2b) and the mean RMSE of individual ensemble members to independently withheld data (12.9±1.1 µatm). The mean standard deviation across all grid cells within the 10-fold ensemble is 2.2±1.3 µatm (mapped in supplementary Figure 3). Each individual ensemble member also performed relatively well compared to the 10% subsample of corresponding 10-fold evaluation data (mean RMSE = 17±2 µatm; supplementary Figure 4).

### 3.2 Uncertainty calculations

Uncertainty in the ANN estimated $pCO_2$ product was calculated following Landschützer et al. (2018, 2014), Roobaert et al. (2019), and Keppler et al. (2020) (Eq. 2), where the overall $pCO_2$ product uncertainty ($\theta_{pCO2}$) is calculated from the square root of the sum of the four squared errors: observational uncertainty ($\theta_{obs}$), gridding uncertainty ($\theta_{grid}$), ANN interpolation uncertainty ($\theta_{map}$), and ANN run randomness uncertainty ($\theta_{run}$).

$$\theta_{pCO2} = \sqrt{\theta_{obs}^2 + \theta_{grid}^2 + \theta_{map}^2 + \theta_{run}^2},$$ (2)




Observational uncertainty ($\theta_{obs}$ = 3.1 µatm) is the measurement uncertainty of $p$CO$_2$ in the field, evaluated as the average of the uncertainty assigned to each data point according to its SOCAT quality control (QC) flag (between 2-5 µatm). Gridding uncertainty ($\theta_{grid}$ = 1.5 µatm) is associated with gridding SOCAT observations into monthly 1/12°x1/12° bins, evaluated as the average standard deviation among $p$CO$_2$ values within each grid cell with at least 3 data points. ANN interpolation uncertainty ($\theta_{map}$ = 11.1 µatm) is uncertainty introduced by interpolating the $p$CO$_2$ observations using the SOM-FFN

approach, evaluated as the RMSE from the ANN ensemble output compared to the independent withheld SOCAT data (Figure 2b). The standard deviation of the ensemble (ensemble spread) gives an indication of how robust our estimate is from one run to the next using different 10-fold training data (Section 2.5; Keppler et al., 2020). ANN run randomness uncertainty ($\theta_{run}$ = 2.2 µatm) comes from the mean standard deviation between 10-fold ensemble members (Section 2.5 & 3.1), which is less than the comparison of each member of the ensemble with the ensemble mean (supplementary Figure 3;

Figure 2c).

Overall product uncertainty combining all four components according to Eq. (2) is 12 µatm, with the contribution of ANN interpolation uncertainty being the largest. Our product uncertainty is comparable to reported open ocean uncertainty values from global products (Landschützer et al., 2014), as well as a regional product in the California Current System (Sharp et al., 2022). Combining the reported uncertainty in the gas transfer velocity (Section 2.6) and the overall $p$CO$_2$ product uncertainty

yields an average uncertainty of ±0.24 mol m$^{-2}$ yr$^{-1}$ in the air-sea gas flux, with the largest fraction of the error stemming from the uncertainty of the gas transfer velocity. The total uncertainty in the flux corresponds to roughly 20% of individual grid cell calculated flux values.

## 3.3 Improvement relative to a global product

The ANN-NEP $p$CO$_2$ product created here shows improved performance over the Landschützer et al. (2020b) global product

at each timestep within the study area when compared to SOCAT data gridded at 1/12°x1/12° (Figure 3), illustrating the importance of regional high-resolution estimates in resolving fine scale variations. Across all evaluation metrics the global product does not perform as well in the region compared to SOCAT training data (RMSE = 14; $r^2$ = 0.74; mean bias = -2; c$_0$ = 0.68; MAE = 10; compared to Figure 2a). This improvement suggests a regional high-resolution product can narrow the range of variability in predictor data within the SOM clustering step and present $p$CO$_2$ observation data with greater

correlation to the FFN. In the Landschützer et al. (2020b) global product, there is often only one SOM biogeochemical province covering the whole region, forcing non-linear relationships in the FFN to be built around greater variability in $p$CO$_2$ observation data from a wider range of geographic areas. The ANN-NEP regionally specific four SOM biogeochemical province grouping could alleviate this shortcoming in the FFN step. The improvement in our high-resolution product is particularly evident in the seasonal amplitude, where differences between ANN-NEP and Landschützer et al. (2020b) exceed

the product uncertainty in 25% of grid cells (supplementary Figure 5a). The largest seasonal amplitude differences occur in the north Alaskan Gyre region, and south of the North Pacific Current (supplementary Figure 5b). The additional spatial





resolution and temporal details in the regional high-resolution product provide key information to inform future observation programs including potential mooring locations. The value added in stepping to a high-resolution regional product proves particularly useful in resolving biogeochemical gradients within the subpolar Alaskan Gyre system in our study area (Section
4).

## 3.4 Performance at coarser resolutions

Stepping to a higher spatial resolution drastically decreases the ratio of gridded $pCO_2$ observations compared to the total number of grid cells (Figure 4f), nevertheless the ANN experiences minimal loss in performance across different spatial resolutions (Figure 4a-e). Globally, most open ocean observation-based $pCO_2$ products interpolate on a 1°x1° gridded
resolution (Landschützer et al., 2020b; Global Ocean Surface Carbon, E.U. Copernicus Marine Service Information MULTIOBS_GLO_BIO_CARBON_SURFACE_REP_015_008; Denvil-Sommer et al., 2019; Zhong et al., 2022), with most coastal or regional products using a 1/4°x1/4° grid cell size (Laruelle et al., 2017; Sharp et al., 2022; Hales et al., 2012; Nakaoka et al., 2013), with a few regional products stepping to even higher resolutions (e.g., 1-km in Chen et al. 2016; 4-km in Parard et al. 2015, 2016; 11-km in Xu et al. 2019). To determine how the network preforms when producing a coarser
resolution product, we tested the same configuration of our tuned 1/12°x1/12° ANN at various resolutions (Figure 4). The predictor variables and SOCAT $pCO_2$ observations were simply bin-averaged to coarser grid cell sizes (i.e., 1°, 1/2°, 1/4°, 1/8°).

Using the same ANN configuration between the different resolutions (i.e., optimal SOM biogeochemical provinces, appropriate predictors, neuron number in the first hidden layer, etc., see Section 2.5), the most important parameter for
reducing overfitting at each resolution becomes the internal data division ratio between the pCO₂ training data used by the ANN to train and internally evaluate (Figure 4). We tested a suite of data division ratios between 99% of data used to train / 1% used to internally evaluate to a 50/50 split at 1% intervals for each resolution (Figure 4). These tests were run without the 10-fold cross-evaluation ensemble approach. To quantify the optimal ratio at each resolution, we used an overfitting metric (Eq. 3) equal to the larger of the training or independently withheld data RMSE, plus the absolute value of the difference
between the two:

$$\text{Overfitting metric} = \max\left(\text{RMSE}_{training}, \text{RMSE}_{withheld}\right) + \left|\text{RMSE}_{training} - \text{RMSE}_{withheld}\right|, \tag{3}$$

Using an internal data division ratio optimized based on the overfitting metric, an ANN interpolated $pCO_2$ product with an uncertainty value of 12.5±0.4 µatm (see Section 3.2; supplementary Table 2) is possible at each of the coarser resolutions (Figure 4a-e; supplementary Table 2). For comparison, the reported uncertainty in a global product (Landschützer et al.,
2014) ranges from 9 to 18 µatm. This finding creates a precedent for stepping to a higher resolution product with nearly no loss in performance, overcoming the overfitting concern with increased resolution (Rosenthal, 2016).





## 4 Air-sea CO₂ fluxes

With the estimated ANN $p$CO₂ product displaying a strong ability to accurately represent regional $p$CO₂ variability in the Northeast Pacific (Section 3), we calculate air-sea CO₂ fluxes in the region (Eq. 1). Long-term (1998-2019) mean $p$CO₂ and

air-sea CO₂ fluxes display similar patterns (Figure 5). In the northwest of our study area, high $p$CO₂ and net CO₂ outgassing to the atmosphere correspond to the influence of the upwelling subpolar Alaskan Gyre system (Figure 5; Figure 1c). Lower $p$CO₂ values and stronger atmospheric CO₂ uptake occur in the North Pacific Current region (Figure 1c) to the south and along the eastern study area margin (Figure 5). The gradient of the gyre captured in the high-resolution estimate improves regional understanding with the largest differences between the Landschützer et al. (2020b) global product occurring in the

north (basin-wide absolute difference 2-5%; supplementary Figure 5). Higher resolution in the gyre gradient also provides regional context to carbon measurements made at the Ocean Station Papa mooring, often used to represent the Alaskan gyre (e.g., Jackson et al. 2009), which is actually situated approximately between the two regions, and along the Line P monitoring program.

### 4.1 Seasonal variability

To determine seasonal cycle drivers, we decompose the climatological $p$CO₂ into a thermal and non-thermal component (Takahashi et al., 2002, 1993):

$$pCO_{2\,(T)} = pCO_{2\,(am)} \times \exp\left[0.0423\left(T_{(mm)} - T_{(am)}\right)\right], \tag{4}$$

$$pCO_{2\,(NT)} = pCO_{2\,(mm)} \times \exp\left[0.0423\left(T_{(am)} - T_{(mm)}\right)\right], \tag{5}$$

$$R_{(T\,NT^{-1})} = \frac{\max\left(pCO_{2\,(T)}\right) - \min\left(pCO_{2\,(T)}\right)}{\max\left(pCO_{2\,(NT)}\right) - \min\left(pCO_{2\,(NT)}\right)}, \tag{6}$$

Here the subscripts $T$ and $NT$ represent thermal and non-thermal effects, respectively, while subscripts $am$ and $mm$ represent annual mean and monthly mean values, respectively. Eq. 4 imposes the empirical temperature dependency on the annual mean $p$CO₂ value providing an estimate of seasonal temperature control (Sarmiento and Gruber, 2006; Takahashi et al., 2002). Eq. 5 removes the temperature dependency from the monthly mean $p$CO₂ values providing an estimate of the residual, non-thermal controls on $p$CO₂ including circulation, mixing, gas exchange, and biology. The ratio of the seasonal

amplitudes of the two components (Eq. 6; $R_{(T\,NT^{-1})}$) can differentiate the dominant process, where a value greater (less) than one indicates that thermal (non-thermal) processes dominate.

Seasonally, the northern Alaskan Gyre region of our study area (latitudes north of 52°N; Figure 6a&b), flips from outgassing in the wintertime to uptake in the summer in the climatological air-sea CO₂ flux (Brady et al., 2019; Palevsky et al., 2013; Chierici et al., 2006). The change in the sign of the flux is driven by a 40 µatm difference between winter maxima and

summer minima $p$CO₂ climatology values (Figure 6b). In the Landschützer et al. (2020a) climatology, this seasonal dipole in the Alaskan Gyre also exists displaying a 40 µatm seasonal $p$CO₂ range. Similar patterns exist in the Takahashi et al. (2014,



2009, 2002) climatologies as well (approximately 45-50 µatm). Increased wind stress curl drives stronger gyre circulation in the fall and winter, upwelling and entraining nutrient and $CO_2$-rich subsurface waters into the surface ocean, increasing the non-thermal $pCO_2$ component (Figure 6b), leading to outgassing (Figure 6a; Chierici et al. 2006). Through the spring and summer, biological drawdown, preconditioned by the upwelled, mixed, and entrained nutrients, decrease the surface ocean non-thermal $pCO_2$ component (Figure 6b; Harrison et al. 1999), enhancing uptake (Figure 6a). Although the seasonal amplitude of the temperature component is also large in the north, these non-thermal controls dominate ($R_{(T \, NT^{-1})} = 0.84$).

In the south part of our study area, the North Pacific Current region (latitudes south of 52°N; Figure 6a&c) acts as a strong $CO_2$ sink through the winter transitioning to a weak sink through the summer. Whereas in the Alaskan Gyre region the seasonal cycle of $pCO_2$ is dominantly controlled by non-thermal drivers (Figure 6b), the North Pacific Current region experiences a near balance between opposing drivers (Figure 6c; $R_{(T \, NT^{-1})} = 1.02$). In the North Pacific Current region, we see a much smaller seasonal amplitude in $pCO_2$ (15 µatm; Figure 6c), peaking in July with warming, falling to a minimum in October. The seasonal amplitude is dampened by the competing effect of temperature changes in solubility, and changes in dissolved inorganic carbon concentration through biological drawdown and changing mixed layer depth (Wong et al., 2010; Sutton et al., 2017). With minimal seasonal variation in seawater $pCO_2$, the seasonal change in atmospheric $CO_2$ uptake south of 52°N (Figure 6a) is dominantly driven by higher wind speed through the winter months (mean increase of 55% over summer climatological values).

## 4.2 Alaskan Gyre upwelling strength

On sub-decadal to decadal timescales, there is a strong correlation between air-sea $CO_2$ flux anomalies and SSH anomalies in the Alaskan Gyre region of our study area ($r^2 = 0.93$, $p < 0.01$; Figure 7b&c; supplementary Figure 6). In this subpolar gyre, prevailing winds cause upwelling driven by Ekman pumping (Gargett, 1991), but the strength varies. During 1998-2002 as well as 2006-2013, we observe strong winter and spring outgassing in the Alaskan Gyre, with flux densities as high as 3.6 mol m$^{-2}$ yr$^{-1}$ in January 2000 (Figure 7a). In these same periods, anomalously low sea level pressure over the Alaskan Gyre led to anomalously strong wind stress curl which enhanced Ekman pumping and depressed SSH (Figure 7b; Mann and Lazier 2006; Hristova et al. 2019). The stronger upwelling brought $CO_2$-rich subsurface water to the surface (Lagerloef et al., 1998). Conversely, during the periods of anomalously high sea level pressures and positive SSH anomalies (2003-2005; 2014-2020; Figure 7c), there is less upwelling of $CO_2$-rich subsurface water to the surface, allowing primary productivity to draw down surface ocean $CO_2$ (McKinley et al., 2006), enhancing $CO_2$ uptake from the atmosphere (Figure 7b).

Our observation-based findings show strong carbon relationships with SSH in the Alaskan Gyre, with correlations between other climate indices being weaker. Over longer timescales, climate-driven regional ocean fluctuations have been shown to modulate the Alaskan Gyre surface water inorganic carbon system (Hauri et al., 2021; Di Lorenzo et al., 2008). The North Pacific Gyre Oscillation and the Pacific Decadal Oscillation indices have both been shown to strongly influence the physics, chemistry, and biology of Gulf of Alaska ecosystem (Di Lorenzo et al., 2008; Newman et al., 2016). Hauri et al. (2021)



showed that the rate of ocean acidification in a hindcast model of the Gulf of Alaska was strongly related to the first
empirical orthogonal function of SSH. We report the same relationship with SSH described in Hauri et al. (2021) as the
dominant control of sub-decadal patterns on air-sea $CO_2$ fluxes from our observation-based $p$CO$_2$ product (Figure 7). Our
estimates of the 12-month running mean air-sea $CO_2$ flux anomaly in the Alaskan Gyre region (Figure 7b) are more weakly
correlated to the North Pacific Gyre Oscillation, Pacific Decadal Oscillation, and the El Niño-Southern Oscillation indices
($r^2$ = 0.63, 0.38, 0.22 respectively; $p$ <0.01). This regional variation in SSH correlating with both observations and models
lends strong evidence for variations in Alaskan Gyre upwelling strength explaining regional biogeochemistry on sub-decadal
to decadal timescales. This finding also highlights the challenges of representing the regional seasonal cycle of the Northeast
Pacific in a climatology within a reference period dominated by one mode of Alaskan Gyre upwelling strength.

### 4.3 Impact of interannual events

On shorter, interannual timescales, basin-wide variability in air-sea $CO_2$ flux is significantly influenced by the impact of
extreme events, with the underlaying sub-decadal and decadal signal amplifying or dampening these impacts. During
persistent marine heatwaves in the Northeast Pacific since 2013, we see strong atmospheric $CO_2$ uptake anomalies fueled by
reduced winter mixing and increased surface density stratification (Figure 8; Bond et al., 2015). The strongest marine
heatwave, known as "the Blob", with sea surface temperature anomalies greater than 3°C or 4 standard deviations above
normal (Freeland and Ross, 2019), persisted in the Northeast Pacific from late 2013 to the end of 2015 driven by an
anomalous high-pressure atmospheric ridge (Bond et al., 2015; Di Lorenzo and Mantua, 2016). The ridge was associated
with a significant decline in local wind speed, decreasing the mixing of deep, colder waters to the surface and raising sea
surface temperatures (Bond et al., 2015; Scannell et al., 2020). The reduced winter mixed layer deepening and associated
limiting of upwelled and entrained nutrient and $CO_2$-rich subsurface waters to the surface has been linked to a relief of ocean
acidification (i.e., anomalously high aragonite saturation states; Mogen et al. 2022). There has also been a reported increase
in net primary production during "the Blob" in both in-situ and satellite records (Long et al., 2021; Yu et al., 2019; Peña et
al., 2019). During "the Blob," we see strong negative air-sea $CO_2$ flux anomalies, particularly in the winter months (October
to December 2014 and 2015), indicative of a 30% increase in uptake relative to climatological monthly means. The increased
atmospheric $CO_2$ uptake is driven by reduced winter wind speeds (by approximately 7%) leading to limited winter mixed
layer deepening, increased surface density stratification, while possibly being enhanced by the increase in net primary
production (Figure 8b).

Through a second marine heatwave from mid-2018 to 2020 (Chen et al., 2021; Amaya et al., 2020; Scannell et al., 2020), we
see a similar magnitude increase in atmospheric $CO_2$ uptake compared to "the Blob" event (Figure 8b). Through some of the
largest SST anomalies (October to December 2018 and 2019) we observed large negative air-sea $CO_2$ flux anomalies
indicating enhanced atmospheric uptake of 45% beyond corresponding climatological monthly means (Figure 8b),
particularly in the Alaskan Gyre (Figure 7a&b). During this marine heatwave, a similar reduction in upper ocean mixing and
limited wintertime entrainment due to reduced wind speed were observed (by approximately 9%; Amaya et al. (2020)) and





resultant reduced surface $pCO_2$ (Franco et al., 2021). Increased net primary production has also been reported (Long et al., 2021). An unusual near-surface freshwater anomaly in the Gulf of Alaska during 2019 contributed to the intensification of the marine heatwave by increasing the near-surface buoyancy and density stratification (Scannell et al., 2020).

Our result that marine heatwaves cause enhanced $CO_2$ uptake in the Northeast Pacific may not be applicable to a wider region. Mignot et al. (2022) described how the impact of marine heatwaves on air-sea $CO_2$ fluxes are the net result of two competing mechanisms: 1) increased sea surface temperatures reducing the solubility of $CO_2$, increasing $pCO_2$ and reducing $CO_2$ uptake, and 2) increased density stratification reducing vertical mixing and entrainment, decreasing surface dissolved inorganic carbon, and increasing $CO_2$ uptake. Their analysis finds that the temperature effect outweighs the advection effect

during persistent marine heatwaves in the whole North Pacific (from 123.5°E to 121.5°W and 23.5°N to 59.5°N; Mignot et al. 2022) reducing $CO_2$ uptake by 29±11%, with the opposite true in the Tropical Pacific (Mignot et al., 2022). However, when looking at our more localized study area in the Northeast Pacific, we find instead that the impact of reduced winter mixing (because of decreased winds and increased density stratification) tipped the balance toward enhanced atmospheric $CO_2$ uptake during these marine heatwaves, again advocating the need for high resolution local studies to better understand

local climate change effects.

Through both "the Blob" and the 2019 marine heatwave, the Alaskan Gyre was in a period of weak upwelling (Figure 7c), leading to a decade-long negative $pCO_2$ anomaly (Figure 8a), in addition to the maximum observed $\Delta pCO_2$ due to the diverging long-term trend with the atmosphere (Section 4.4). Unravelling the individual influence of these interconnected drivers (i.e., marine heatwaves, sub-decadal variability, and long-term trend) is not possible with this product but does

prompt future inquiry.

We do not observe a large change in atmospheric $CO_2$ uptake associated with the 2008 basin-wide ocean iron fertilization event. In August 2008, the eruption of Kasatochi volcano in the Aleutian Islands, Alaska, USA dispersed volcanic ash over an unusually large area of the subarctic Northeast Pacific fueling a massive phytoplankton bloom in the iron-limited region (Langmann et al., 2010; Hamme et al., 2010). Hamme et al. (2010) reported that enhanced biological uptake drew down

$pCO_2$ by approximately 25 μatm at Ocean Station Papa. Basin-wide, we see a decrease of 20 μatm from July to August 2008 in the detrended, deseasonalized ANN $pCO_2$ following the eruption (Figure 8a) with a drawdown of 30 μatm at Ocean Station Papa. The neural network approach does display a tendency to slightly overestimate relatively low $pCO_2$ values (Figure 2a). Because this basin-wide enhanced primary production and surface ocean $pCO_2$ decrease lasted only two months, its impact on the air-sea $CO_2$ flux was limited (Figure 8b). The limited impact could be tied to weaker summer wind speeds

and longer equilibration times (Jones et al., 2014). The eruption occurred during a period of enhanced Alaskan Gyre upwelling (Figure 7c), meaning the event was overlaid on top of an already sub-decade long positive $pCO_2$ anomaly (Figure 8a) perhaps dampening the event's impact. Unfortunately, the lack of direct $pCO_2$ measurements in SOCAT during this time prevents us from further investigating the underlaying causes.





## 4.4 Air-sea CO$_2$ flux trend

Overall, the Northeast Pacific Ocean CO$_2$ sink has become more negative (i.e., become a larger sink; Figure 9b) from 1998 to 2020 at a rate of -0.043±0.004 mol m$^{-2}$ yr$^{-2}$. Looking at the start and end of the timeseries, the average flux from 1998 to 2002 appeared to be a small atmospheric CO$_2$ sink at -0.7±0.6 mol m$^{-2}$ yr$^{-1}$, compared to the sink from 2016-2020 at -1.6±0.8 mol m$^{-2}$ yr$^{-1}$. Regionally, we don't see a statistically significant trend in the satellite-based ocean surface wind speed data over this time ($p$ >0.1; Mears et al. 2019). However, the timeseries endpoints are both representative of different Alaskan

Gyre upwelling modes (Figure 7c), with the timeseries starting in a sub-decade long positive $p$CO$_2$ anomaly and ending during a decade long negative $p$CO$_2$ anomaly. Decadal trends will be sensitive to the start and end point of the timeseries (e.g., Fay and McKinley 2013). We caution that our trend results may not be representative of longer time periods (i.e., from industrial onset).

Taking the full study area deseasonalized (Section 2.4), area-averaged $p$CO$_2$, we calculated trends based on shorter time
series within our data using different monthly timeseries start and end dates (Figure 10). Based on $p$CO$_2$ data timeseries ranges greater than 10 years (between 1998 and 2020), 87% of trends are less than the atmospheric trend with a mean of 1.59±0.27 µatm yr$^{-1}$ (N = 9222 at a monthly timestep; Figure 10). In the remaining 13% of total timeseries start and end date combinations, there is a pronounced very steep trend exceeding the atmospheric. Date combinations leading to trends greater than atmospheric could be partially tied to if both the start and end dates fall within a strong Alaskan Gyre upwelling mode driving positive $p$CO$_2$ anomalies. However, the Alaskan Gyre region makes up only about 25% of the total study area

(region north of 52°N; Section 4.2), and trends in Figure 10 represent the ANN-NEP full spatial domain.

The rate of change in the air-sea CO$_2$ flux over the study period is largely due to the increasing gradient with the atmosphere (Figure 9a). Over the full study area from 1998-2020, the ANN-NEP $p$CO$_2$ trend is 1.4±0.1 µatm yr$^{-1}$. The Landschützer et al. (2020b) global product trend in the region is similar at 1.5±0.1 µatm yr$^{-1}$. At Ocean Station Papa, the ANN-NEP $p$CO$_2$

trend is 1.5±0.1 µatm yr$^{-1}$, in agreement with the observed trend based on discrete samples collected 1–3 times per year (1.6±0.8 µatm yr$^{-1}$ between 1990-2019; Franco et al. 2021). The ocean $p$CO$_2$ trend is not as rapid as the atmospheric increase of 2.12±0.03 µatm yr$^{-1}$ over the same period (Figure 9a). Sutton et al. (2017) also reported a lag with the atmosphere at Ocean Station Papa with a Δ$p$CO$_2$ trend of -1.5±0.9 µatm yr$^{-1}$ from the 2007-2014 mooring $p$CO$_2$ data. The ANN-NEP Δ$p$CO$_2$ trend at Ocean Station Papa is -0.67±0.05 µatm yr$^{-1}$.

The observed lag in the increase in surface ocean $p$CO$_2$ with respect to atmospheric $p$CO$_2$, causing an increasing air-sea gradient (Δ$p$CO$_2$), may be attributed to interaction with subsurface water. We find a strong spatial correlation between the trend in Δ$p$CO$_2$ and the calculated average vertical velocity associated with Ekman pumping calculated from zonal and meridional wind speeds ($r^2$ = 0.64, $p$ <0.01; Figure 11b; Mears et al., 2019). Fay and McKinley (2013) describe regions impacted by upwelling from depth having shallower $p$CO$_2$ trends and greater divergence with the atmosphere based on

models and observations. Dissolved inorganic carbon increases with depth, causing enhanced vertical mixing to increase surface ocean $p$CO$_2$ over the seasonal cycle (Sections 4.1 to 4.3). However, in the long-term dissolved inorganic carbon is





increasing most in surface waters, due to direct uptake of atmospheric $CO_2$, and least at depth. The supply to the surface of subsurface waters with low anthropogenic carbon causes a lag in the rate of increase in surface ocean $pCO_2$. The anthropogenic carbon signal in the intermediate to deep waters in this region are some of the smallest in the global ocean due
to circulation patterns (Sabine et al., 2004; Gruber et al., 2019; Carter et al., 2019; Clement and Gruber, 2018). Regions within our study area with greater connection between surface and deep waters, such as the center of the Alaskan Gyre in the north (Van Scoy et al., 1991), are experiencing the largest divergence with the atmosphere. With a joint increase in projected future wind speeds  (Zheng et al., 2016; Young and Ribal, 2019; Wanninkhof and Triñanes, 2017), and a growing $\Delta pCO_2$, the region is likely to become a stronger net annual sink for atmospheric $CO_2$.

**5 Conclusions**

Using a high-resolution regional neural network approach, we represent $pCO_2$ measurement variability well in the Northeast Pacific Ocean. We interpolated sparse observations using non-linear relationships developed with a neural network approach based on predictor data from satellite and reanalysis products to create a continuous monthly $pCO_2$ estimate at 1/12°x1/12° spatial resolution. Using a cross-evaluation ensemble approach we were able to produce a robust $pCO_2$ product that
represents regional variability with an uncertainty of 12 µatm. We found that stepping to a significantly higher spatial resolution regional $pCO_2$ product led to nearly no loss in performance despite a much lower ratio of gridded $pCO_2$ observations compared to the total number of grid cells. The most important parameter for reducing overfitting across regional $pCO_2$ products with different spatial resolutions was the internal division of training data. Higher resolution products require more direct training data and less data to internally evaluate, while still comparing to independent withheld
data. This work shows that high-resolution, high-performance, observation-based neural network derived $pCO_2$ products can be developed when reducing the complexity of controlling processes by focusing on specific regions. However, chosen predictor variables need to be regionally specific considering "process focused" influences on the local carbon system. Our reported optimization of the internal data division ratio between network training and evaluation data indicates the importance of this choice when moving to a higher spatial resolution. Increased spatial resolution will be necessary to
capture variability in regions strongly influenced by mesoscale processes, enabling resolution of oceanographic features such as eddies, upwelling regimes, and gyre system gradients.

We report pronounced variability in marine $CO_2$ uptake in the Northeast Pacific Ocean dominantly driven by the control of Alaskan Gyre upwelling and connectivity to subsurface waters. Overall, the open ocean Northeast Pacific acted as a net sink for atmospheric $CO_2$ from 1998 to 2020 with an average basin wide air-sea $CO_2$ flux of -1.2±1.4 mol m$^{-2}$ yr$^{-1}$ but with
pronounced seasonality. In the northern Alaskan Gyre region, wintertime upwelling and entrainment lead to significant outgassing. In the southern North Pacific Current region, the seasonal flux cycle is largely driven by wind speed where the seasonal change in surface ocean $pCO_2$ remains small. Based on our product, upwelling strength of the Alaskan Gyre dominates air-sea $CO_2$ flux variability in that region on sub-decadal to decadal timescales. During prolonged periods of





enhanced gyre upwelling, we see strong winter outgassing driven by upwelled and entrained $CO_2$-rich subsurface waters.
During periods of weak gyre upwelling, the northern part of our study area acts as a sink for atmospheric $CO_2$ year-round.
During two recent marine heatwaves we see enhanced $CO_2$ uptake due to limited wintertime entrainment of subsurface
waters resulting from weaker winds. However, we observed minimal impact on atmospheric $CO_2$ uptake following a 2008
volcanic eruption, with air-sea $CO_2$ flux anomalies linked to enhanced biological uptake via iron fertilization lasting only
two months. The gradient between the Northeast Pacific surface ocean $pCO_2$ and atmospheric $CO_2$ is increasing, pushing the
region towards becoming an enhanced sink for atmospheric $CO_2$. We see the largest increase in the gradient, and so potential
for greater future uptake, at the center of the Alaskan Gyre where, through upwelling, there is a strong connection with
subsurface waters low in anthropogenic $CO_2$.

Our analysis illustrates the complex interplay between factors driving air-sea $CO_2$ flux variability at varying temporal scales
across the study domain and within broad subregions (Alaskan Gyre and North Pacific Current regions) allowing us to
suggest what resources will be needed to make further advances. Improvement of estimated $pCO_2$ would benefit from an
increase in the number of $pCO_2$ observations used for training. We recommend prioritizing additional measurements in the
northern Alaskan Gyre region in future observational programs. Our estimated fluxes in the gyre are large (both uptake and
outgassing), but observations are sparse, leading to the largest standard deviations between our cross-evaluation ensemble
members (supplementary Figure 3). The impact of sub-decadal to decadal variability on the trend in surface ocean $pCO_2$ and
in regional atmospheric $CO_2$ uptake emphasises the importance of long duration timeseries sites and programs to capture the
natural cycles of variability and accurately estimate change. Our findings and estimated $pCO_2$ product serve as
environmental baselines, which could be used to inform future marine carbon dioxide removal in the Northeast Pacific at the
basin and regional scale. However, use of our product at the individual grid cell level is not encouraged as errors likely
remain high, whereas over broader regions these errors average away. Our study serves as an important initial step in
creating a complete carbon budget for the Northeast Pacific, with coastal, pelagic, and benthic carbon stocks and fluxes still
to be resolved.

**Code and data availability**

All data used is publicly available. ANN-NEP $pCO_2$ and air-sea $CO_2$ flux fields are available through the National Center for
Environmental Information Ocean Carbon and Acidification Data System (NCEI OCADS Record ID: BGSH2HNRP). $pCO_2$
data are from the Surface Ocean $CO_2$ Atlas (SOCAT) v2021 (available at https://www.socat.info/) as well as additional data
from Fisheries and Oceans Canada February 2019 Line P cruise (available at https://www.waterproperties.ca/linep/). Sea
surface temperature and chlorophyll-$a$ are from the European Space Agency Climate Change Initiative (available at
https://climate.esa.int/en/odp/#/dashboard). Sea surface salinity, sea surface height, and mixed layer depth are from
Copernicus        Marine        Environment        Monitoring        Service        (available        at
https://data.marine.copernicus.eu/product/GLOBAL_MULTIYEAR_PHY_001_030/description). Ocean surface wind data

are from Cross-Calibrated Multiplatform version 2 Wind Vector Analysis Product (available at https://www.remss.com/measurements/ccmp/).

## Author contributions

PD and PL developed the neural network code and created the product with help from RH, DI, NS, and MA. PD, RH, DI, and PL contributed to the interpretation and analysis of the results. All co-authors contributed to editing the manuscript. RH and DI supervised the project work. PC provided data and consultation. PD prepared the manuscript with contributions from all co-authors.

## Competing interests

The contact author has declared that none of the authors has any competing interests.

## Disclaimer

This article reflects only the authors' view – the funding agencies as well as their executive agencies are not responsible for any use that may be made of the information that the article contains.

## Acknowledgement

Ocean Station Papa mooring timeseries site, plus the Line P program are operated by the National Oceanic and Atmospheric Administration and Fisheries and Oceans Canada.

## Financial support

Funding for this project was provided by the Natural Sciences and Engineering Research Council of Canada (NSERC) through the Advancing Climate Change Science in Canada program (grant# ACCPJ 536173-18) to RH. Funding from Fisheries and Oceans Canada's Aquatic Climate Change Adaptation Service Program to PC supported the analysis of recent underway $pCO_2$ measurements made by the Line-P program (grant# 96036). PD financial support also provided by a Natural Sciences and Engineering Research Council of Canada (NSERC) Doctoral Postgraduate Scholarship.

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



**Table 1 Northeast Pacific Open Ocean artificial neural network predictor variables, and their corresponding source, original temporal and spatial resoltuions, and processing steps used for this study.**

| Predictor variable | Source | Original resolution | | Processing |
|---|---|---|---|---|
| | | **Temporal** | **Spatial** | |
| *Satellite-based product* | | | | |
| Sea surface temperature (SST) | SST_cci Level 4 Analysis Version 2.1 | Daily | 1/20°x1/20° | Averaged to monthly |
| Chlorophyll-*a* (Chl) | Ocean_Colour_cci Version 5.0 | Daily | 1/24°x1/24° | Averaged to monthly, aggregated to 1/12°x1/12°, log10-transformed |
| *Satellite and in-situ observation data assimilated reanalysis product* | | | | |
| Sea surface salinity (SSS) | Copernicus Marine Service Global Reanalysis PHY_001_030 | Monthly | 1/12°x1/12° | None |
| Sea surface height (SSH) | | | | None |
| Mixed layer depth (MLD) | | | | log10-transformed |
| *Atmospheric-measurement-based interpolation product* | | | | |
| Atmospheric $\chi CO_2$ | Landschützer et al. (2020) - NCEI Accession 0160558 | Monthly | 1°x1° | Interpolated to 1/12°x1/12° |



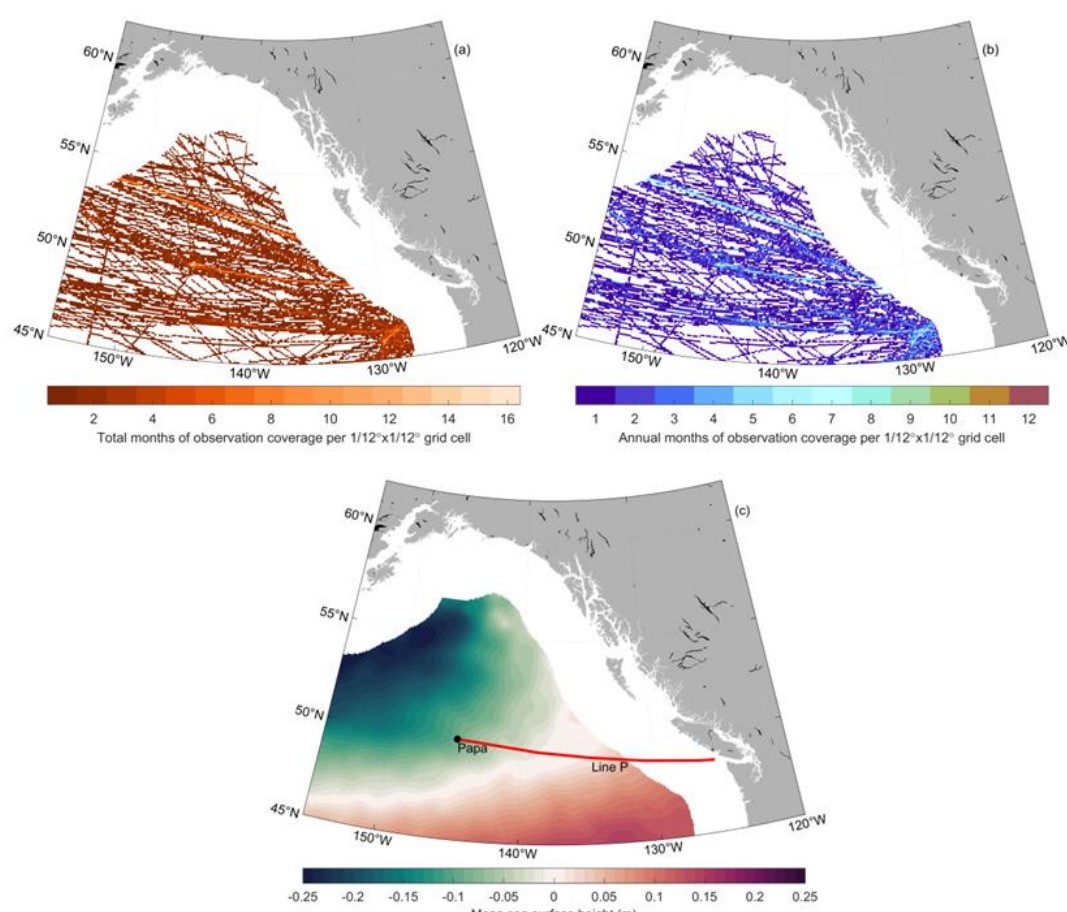

**Figure 1 (a) Total number of months of observational coverage from Surface Ocean CO₂ Atlas (SOCAT) v2021 (Bakker et al., 2016) and additional data from Fisheries and Oceans Canada February 2019 Line P cruise (https://www.waterproperties.ca/linep/) per 1/12°x1/12° grid cell. (b) Number of unique annual months of observational coverage per 1/12°x1/12° grid cell. (c) Mean sea surface height (SSH; Table 1) shows relative location of the subpolar Alaskan Gyre (negative SSH values), and the North Pacific Current (SSH approximately equal to zero). Ocean Station Papa is labeled and marked with a black circle while Line P is labelled and marked with a red line.**

850



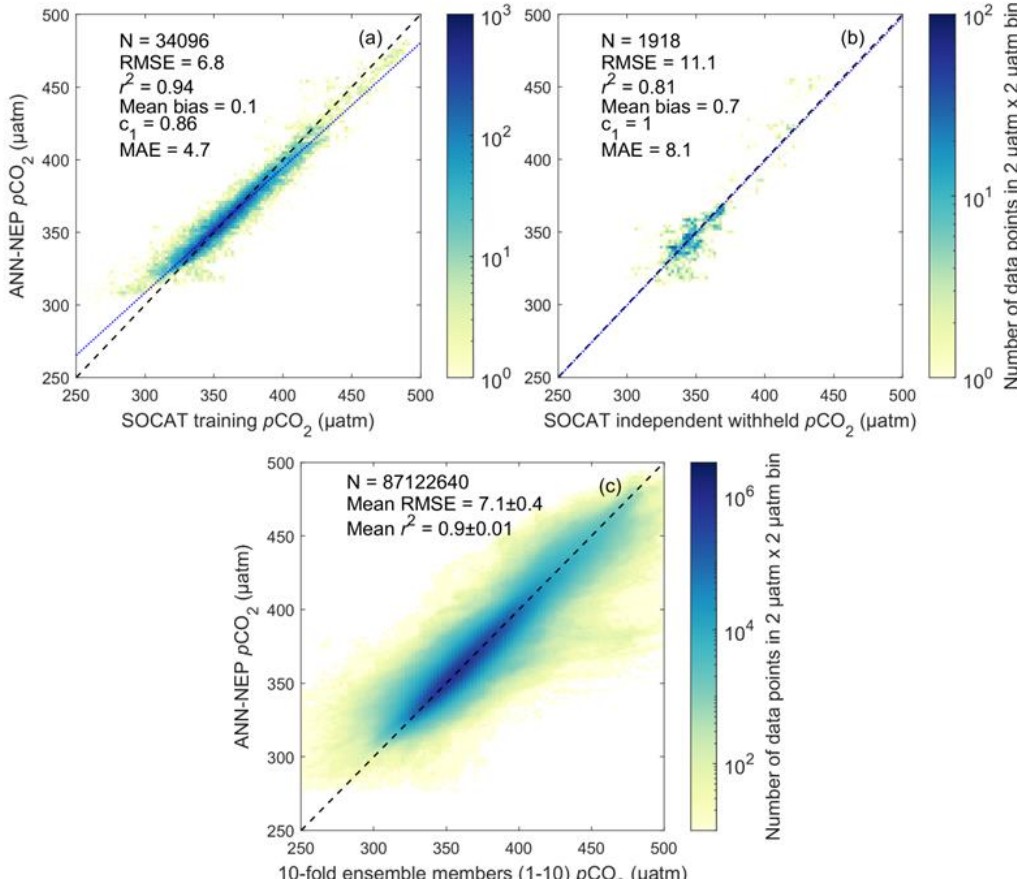

**Figure 2 Regional high-resolution artificial neural network Northeast Pacific (ANN-NEP) ensemble mean pCO₂ against (a) training pCO₂ observation data, and (b) independent withheld pCO₂ observation data. Number of observations (N), root mean squared error (RMSE), coefficient of determination (r²), mean absolute error (MAE), mean bias (calculated as the mean residual), and the slope of the linear regression (c₁). The observed linear relationship is represented by the dotted blue line. (c) ANN-NEP pCO₂ (ensemble mean) against individual ensemble member estimates. Total number of observations (N) across all 10-fold ensemble members (see Section 2.5). Across all panels data are binned into 2 µatm by 2 µatm bins. The dashed black line represents a perfect fit of slope (c₁) = 1 and intercept = 0. Colorbar shows data density on a log scale. Note the order of magnitude difference in the colorbar scale between panels.**



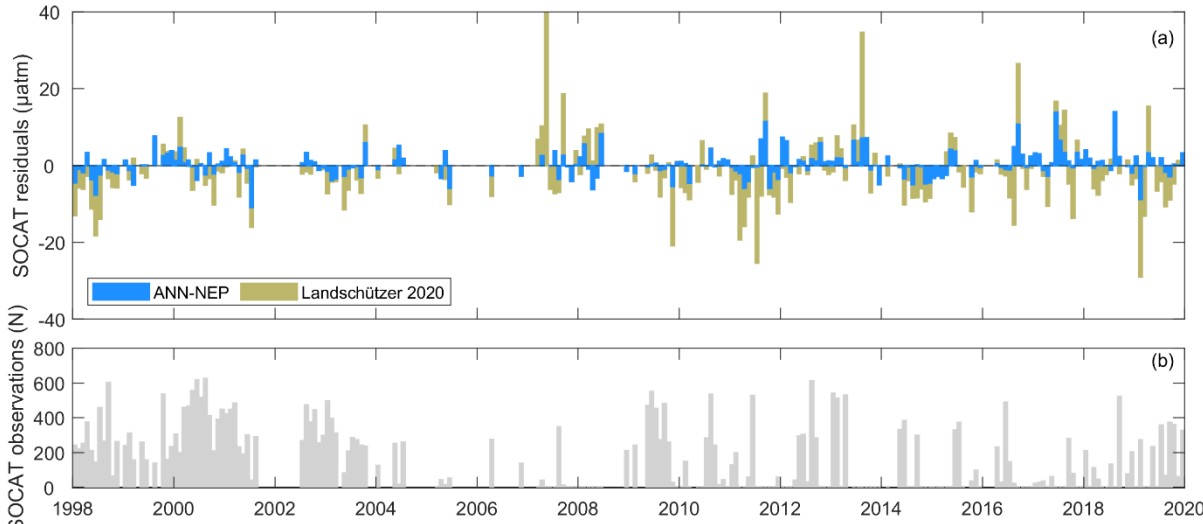

**Figure 3 (a) Mean residuals over the full study area at each timestep of the ANN-NEP pCO₂ estimate in this study and the**
865 **Landschützer et al. (2020b) product re-gridded to 1/12°x1/12° from the gridded SOCAT data. (b) total number of gridded SOCAT**
**observations across the study area at each timestep.**



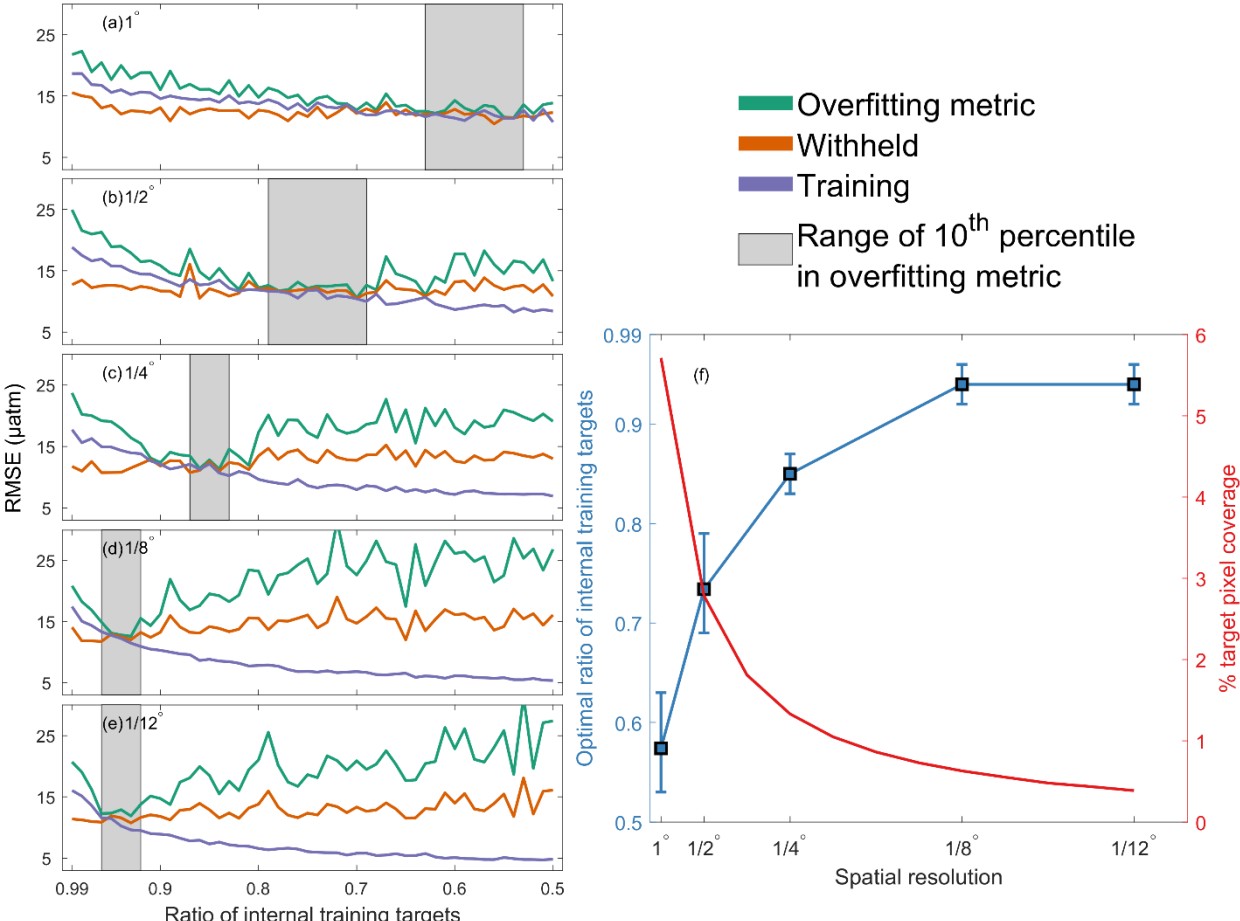

**Figure 4 Varying spatial resolution: (a) 1°, (b) 1/2°, (c) 1/4°, (d) 1/8°, and (e) 1/12° ANN pCO₂ product performance evaluated by the mean RMSE (Section 2.3) of training data (blue line), independently withheld data (orange line), and an overfitting metric (green line; Section 3.4) against internal data division ratios between the pCO₂ training data used by the ANN to train and internally evaluate. The ratios in grey show the range of the lower 10th percentile (5 of 50 runs) of overfitting metric values for each resolution. (f) At each spatial resolution, the lefthand y-axis shows the optimal internal data division ratio with error bars representing the lower 10th percentile of overfitting metric values (same as grey ranges in (a) to (e) with all resolutions converging around RMSE = 12.8±0.4 μatm). The righthand y-axis shows the percent of gridded pCO₂ observations (targets) compared to the total number of grid cells.**



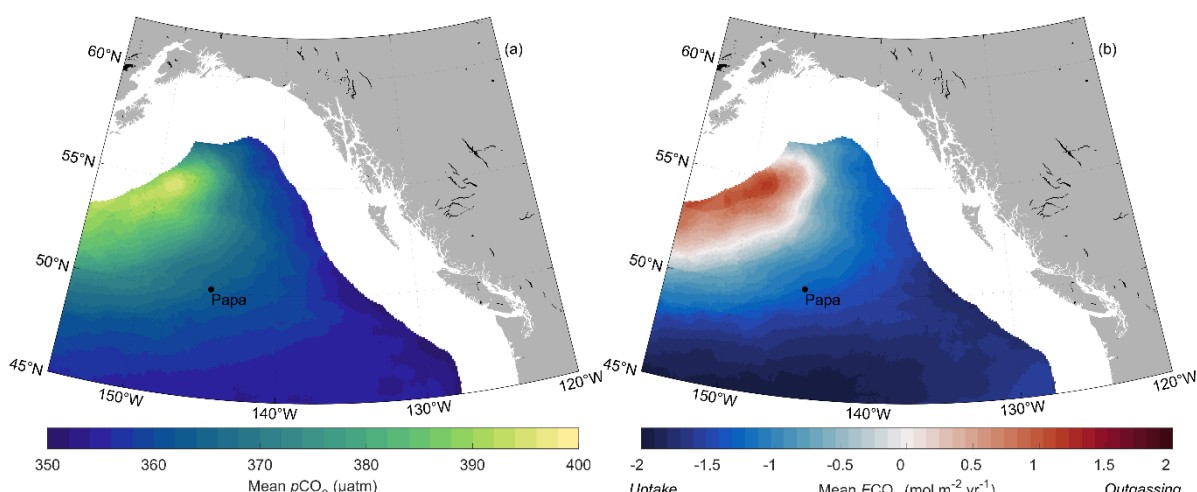

**Figure 5 (a) Long-term (1998-2019) mean ANN-NEP pCO₂ and (b) CO₂ flux density in mol m⁻² yr⁻¹ for the open ocean Northeast Pacific. Negative (positive) flux values indicate CO₂ uptake (outgassing) by the ocean. Ocean Station Papa is shown for reference.**





**Figure 6 (a) Zonally averaged air-sea CO₂ flux from the ANN estimated pCO₂ product climatology along each 1/12° latitude band in the study area plotted against the climatological month along the x-axis (Hovmöller diagram). Negative (positive) flux values indicate CO₂ uptake (outgassing) by the ocean. The dashed black line subdivides the Alaskan Gyre and North Pacific Current regions in the North/South with different seasonal drivers summarized in panels below. (b) Alaskan Gyre region (latitudes north of 52°N) & (c) North Pacific Current region (latitudes south of 52°N) area averaged monthly climatological pCO₂ (solid blue line), thermal component (i.e., changes due to temperature; Eq. 4; dotted red line), and non-thermal component (i.e., changes due to circulation, mixing, gas exchange, and biology; Eq. 5; dot-dash green line). The climatology is plotted over 19 months to emphasize the seasonal cycle.**



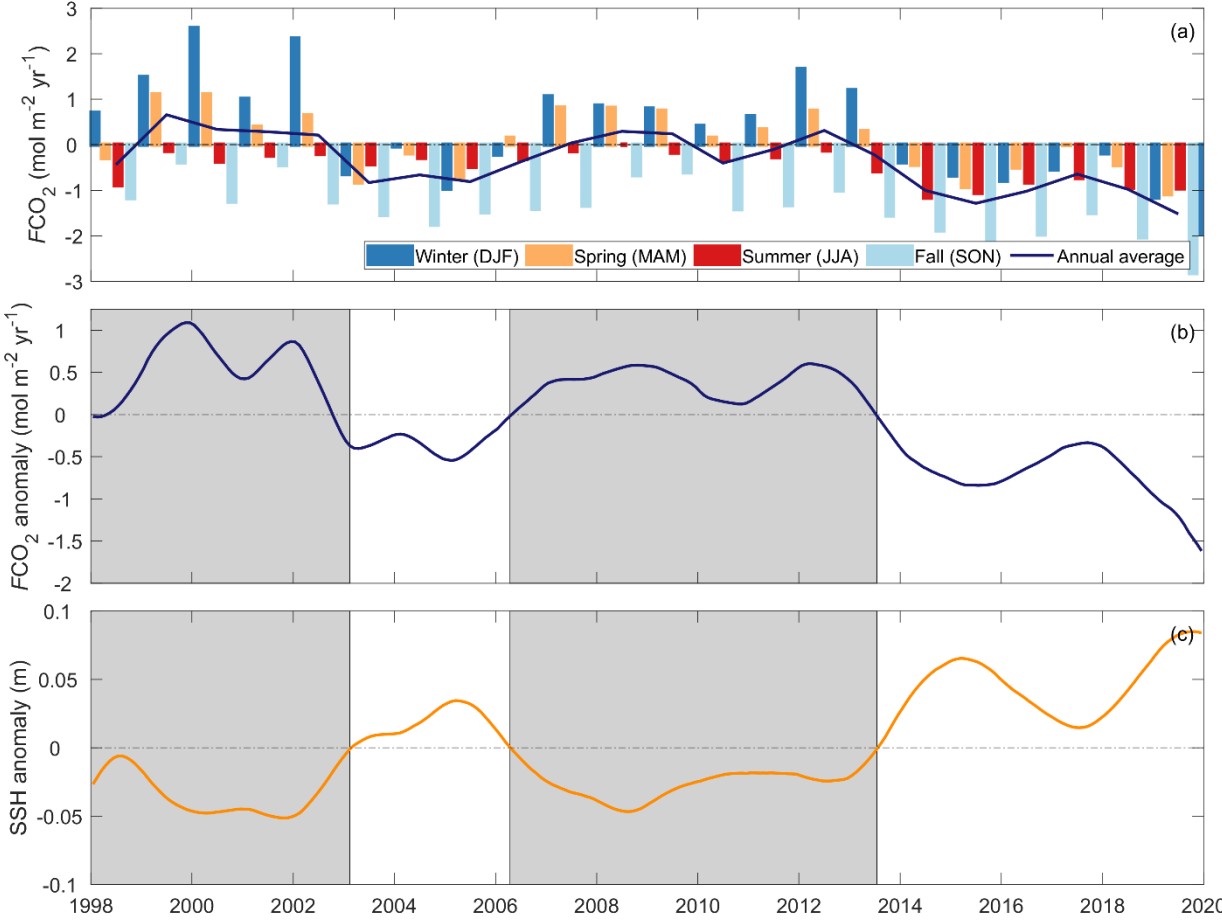

**Figure 7 Alaskan Gyre region of our study area (latitudes north of 52° N). (a) Air-sea CO₂ fluxes grouped by seasonal three-month bins along with the annual average (black line). (b) Air-sea CO2 flux anomalies removing the seasonal cycle (Section 2.4) and applying a 12-month running mean. (c) Sea surface height (SSH) anomalies in the same region removing the seasonal cycle and applying a 12-month running mean. Grey boxes highlight periods of anomalously high Alaskan Gyre upwelling strength corresponding to negative SSH anomalies. Horizontal dashed lines mark zero in each panel. Seasonal groupings in (a) are winter (December, January, February), spring (March, April, May), summer (June, July, August), fall (September, October, November).**

890



**Figure 8** Full study area-averaged interannual variability in (a) pCO₂ anomaly removing the seasonal cycle (Section 2.4) and long-term trend (Section 4.4), (b) air-sea CO₂ flux anomaly, (c) sea surface temperature anomaly, and (d) chlorophyll-a anomaly all removing the seasonal cycle. Grey boxes highlight large interannual events including "The Blob" marine heatwave 2014-2016, a second marine heatwave 2018-2020 ('18 MHW), and a 2008 ocean iron fertilization event following the Kasatochi volcanic eruption (Kasatochi). Horizontal dashed lines mark zero in each panel.



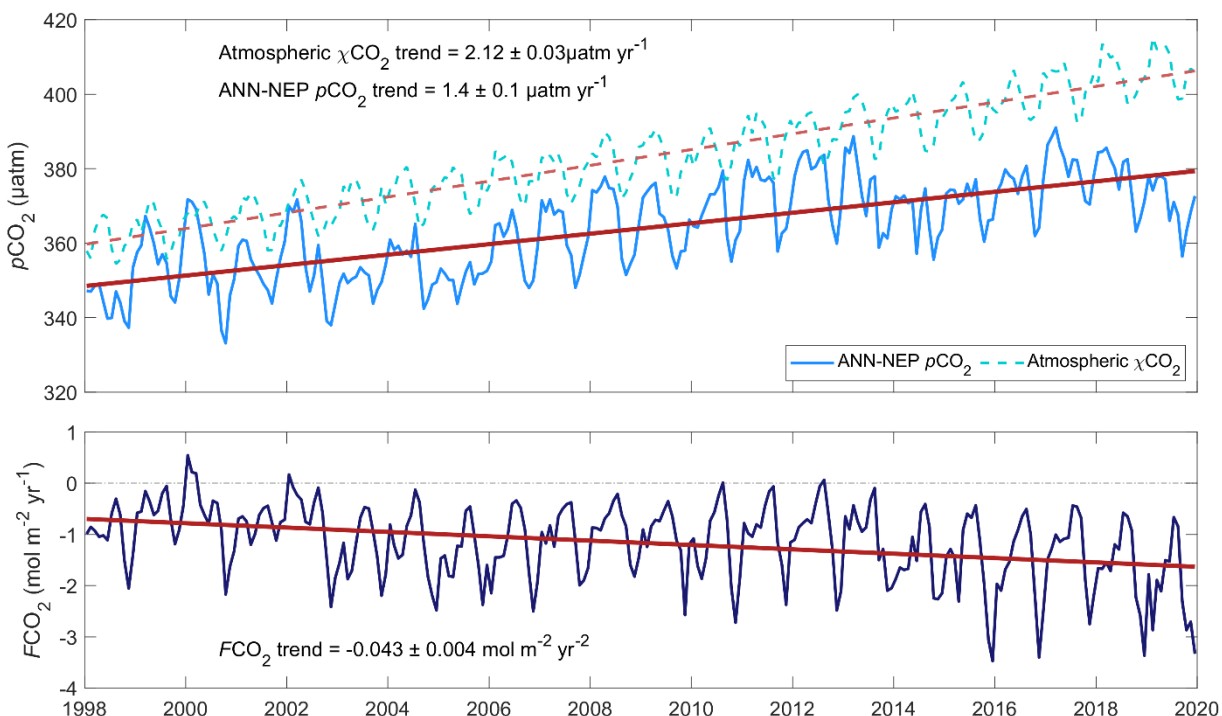

**Figure 9 Full study area-averaged long-term trends in (a) ANN-NEP pCO₂ (solid line) and atmospheric xCO₂ (dashed line), and (b) air-sea CO₂ flux.**



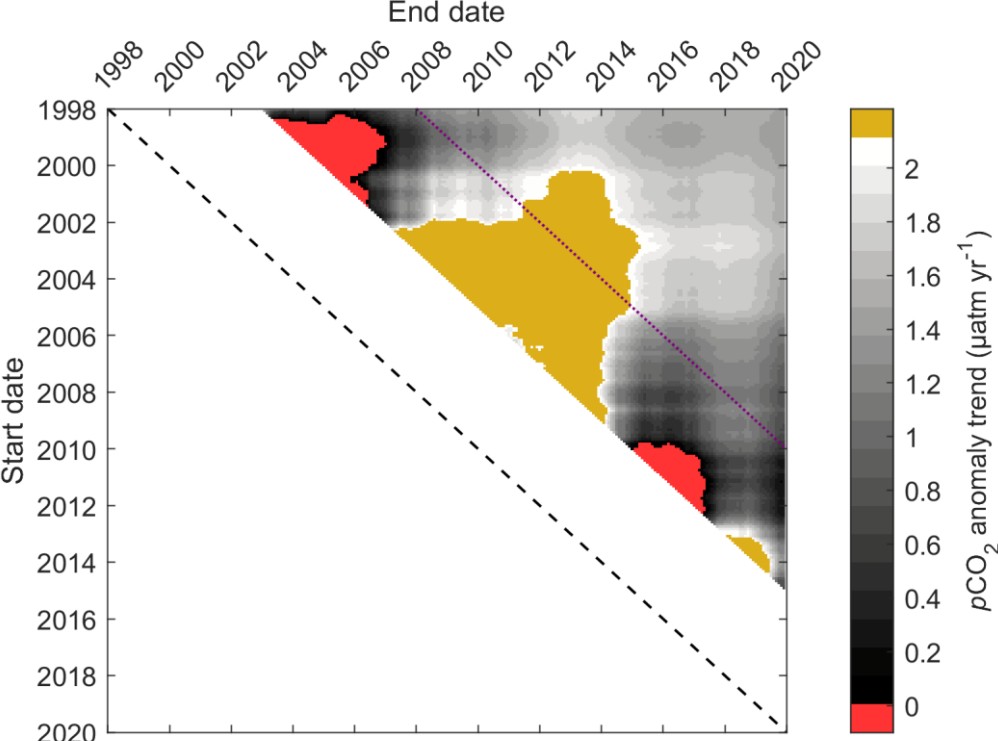

**Figure 10 Full study area-averaged pCO₂ anomaly (removing the seasonal cycle; Section 2.4) linear trend calculated using different monthly timeseries endpoints. Timeseries start from dates on the left and end on a date along the top. The dashed black line indicates equal start and end dates. Trend values are only shown for timeseries of at least a 5-year duration. Red values represent negative pCO₂ trends, gold values represent trends greater than the atmospheric increase (2.12±0.01 µatm yr⁻¹). The purple dotted line indicates a 10-year timeseries duration.**





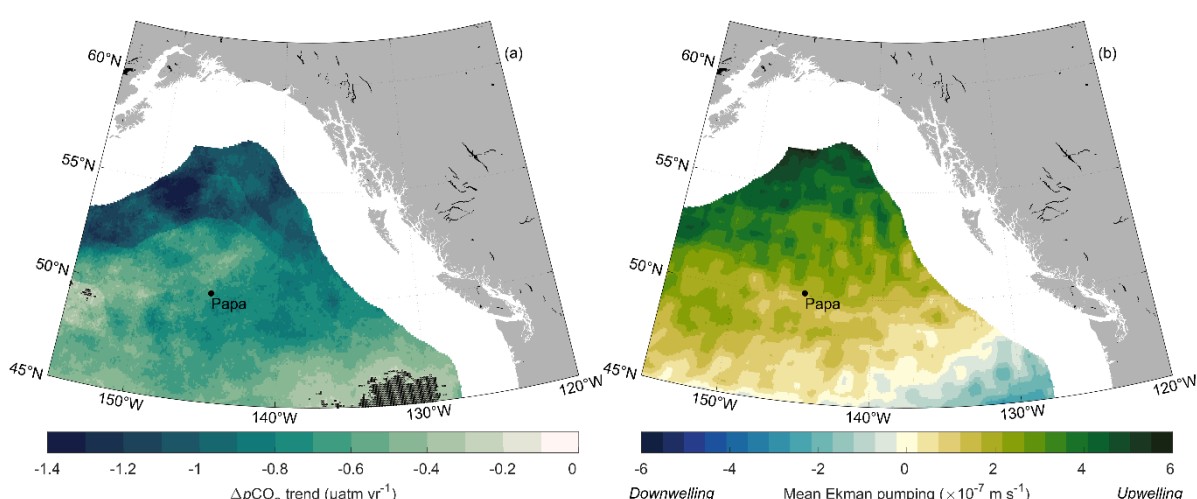

**Figure 11 (a) Trend in ΔpCO₂ where more negative (darker) values indicate an increasing gradient with the atmosphere and a lag in the pCO₂ increase in the surface ocean. Black crosshatches show grid cells with an insignificant calculated trend (outside the**
915 **95% confidence level; p ≥0.05). (b) Calculated average vertical velocity associated with Ekman pumping (calculated from zonal and meridional wind speed) where negative (blue) values indicate downwelling and positive (green) values indicate upwelling. Ocean Station Papa is shown for reference.**