# Peer review of "Estimating Marine Carbon Uptake in the Northeast Pacific Using a Neural Network Approach"

_EGUsphere, 2023_

## Author Comment (AC1)

**Duke et al. Reply on RC1: 'Comment on egusphere-2023-870', Marine Fourrier**

General comments

This paper by Duke and co-authors describes a neural network approach to interpolate observations into a gridded pCO2 data product over the period 1998-2019 at 1/12° resolution specifically in the Northeast Pacific. After presenting the specific circulation and hydrographic features of their area, the authors detail the methodological choices they made in developing the NN and advise the reader on specific training steps to improve performance. They then go on to discuss the wider scientific applications of such a product.

Overall, the paper is very well written, nicely structured and easy to read. The figures could be larger to improve readability, but figures and legends are still clear. The level of detail in the methodological and NN development sections is much appreciated, as it is rare to find such detail in oceanographic research papers. In particular, the decision not to normalise all your inputs and to use dynamic provinces is very interesting. Also all the developments you have done on the internal division data ratio, which could be applied to many other uses.

Thank you Marine Fourrier for your time and careful consideration in the assessment of our manuscript. We are glad that the detailed description of the NN approach and transferability of the method development were received well. Below we present a point-by-point response to comments. Our responses are in blue, with manuscript text in quotations and added/revised text italicised.

A few specific comments remain, as detailed below:

Specific comments

It would be nice to further highlight the potential uses of your data product and/or your method outside the traditional straightforward pCO2/FCO2 observing community (i.e. modellers, determination of climate indices, ...).

We have added the following paragraph to section 5 line 483:

*"The regional, high-resolution pCO2 product created here could serves as a valuable baseline for regional models (e.g., Pilcher et al., 2018; Hauri et al., 2020). The pCO2 product, and associated air-sea CO2 flux estimates, offers continuous coverage in sparsely sampled regions informed by patterns in well sampled neighbouring waters. The product could be used to aid in model evaluation, use in data assimilation, constrain initial conditions, enhance carbon flux process understanding, and improve regional climate change projections."*

As well as additional line to section 4.2 line 351:

*"This relationship supports work showing that the SSH anomaly is an important climate index for the region (Cummins et al., 2005; Di Lorenzo et al., 2008)."*

Have you considered comparing your FCO2 with other products: e.g. SeaFlux (Fay et al., 2021)?

Compared to the averaged SeaFlux $FCO_2$ product (across 6 different observation-based products, with 5 different wind speed products) our estimated $FCO_2$ looks similar outside the higher range of outgassing values (positive fluxes > 5 mol m$^{-2}$ yr$^{-1}$) which exist within the Alaskan Gyre (Figure 5b & 6a).

This comparison has been added in text to section 4 line 290:

*"Our ANN-NEP calculated fluxes compare well to air-sea $CO_2$ fluxes averaged across six unique courser resolution, global observation-based pCO2 products, each using five different wind speed products ($r^2$=0.81; Fay et al. 2021). However, our work suggests that the global product ensemble may underestimate the outgassing signal from the subpolar Alaskan Gyre (Figure 5b; supplemental Figure 7)."*

and included as a new supplemental figure:

[Figure]

*Figure S7 Property to property plot of air-sea $CO_2$ flux density values calculated from ANN-NEP and from SeaFlux v2021.04 (Fay et al., 2021). The SeaFlux estimates have been interpolated to the $1/12\degree$x$1/12\degree$ grid of this study. Number of overlapping grid cells withing the study area (N), root mean squared error (RMSE), coefficient of determination ($r^2$), mean absolute error (MAE), mean bias (calculated as the mean residual), and the slope of the linear regression ($c_1$). The observed linear relationship is represented by the dotted blue line. Data is binned into 0.1 by 0.1 mol $m^{-2}$ $yr^{-1}$ bins. The dashed black line represents a perfect fit of slope ($c_1$) = 1 and intercept = 0. Colorbar shows data density on a log scale.*

End of section 2.1 and Table 1: there seems to be some confusion here about what you have used for what. You mention the xCO2 data produced by NOAA, but in Table 1 you cite Lanschutzer? As in the next sentence you mention the pCO2 climatology by the same authors, restructuring the sentences if this is not a confusion would be very useful.

We have added to the description at the end of section 2.1 line 112:

*"Atmospheric $pCO_2$ in µatm was downloaded from Landschützer et al. (2020), derived from the National Oceanic and Atmospheric Administration Earth System Research Global Monitoring Laboratory (https://gml.noaa.gov/ccgg/globalview/) atmospheric mole fraction of $CO_2$ ($\chi CO_2$) and SST (Reynolds et al. 2002) as well as sea level pressure (Kalnay et al. 1996) following Dickson et al. (2007)."*

This text reflects consistency with the table and alleviates confusion later when in text when $\chi CO_2$ should just be referred to as atmospheric $pCO_2$ in µatm. This was a mistake in the submitted manuscript. Thanks for catching the error.

Table 1: for SST you kept the 1/20° resolution and did not aggregate to 1/12°, unlike the others?

We do aggregate to 1/12°. Table 1 has been corrected to reflect this. Thanks for catching the mistake.

Figure 2c: Is your validation data set representative enough? It does not cover your whole range?

Keeping the independent withheld data randomly selected, yet also representative of the full domain was difficult. The end ranges of the training data (high and low $pCO_2$ values) are rather scarce in the observational data set, making them important for training. Striking the balance between withholding representative data yet showing critical training data can be tricky. Currently, we are unaware of any community-based recommendations that deal with this issue. However, our final estimate of product uncertainty is not based on the training data but rather the independent withheld data, unlike may other observation-based products.

End of section 2.5: You average the outputs of the 10 NN. You do this directly, but have you tried to give the median +/std as this is also useful information.

The ensemble median looked very similar to the mean (property to property plot below).

[Figure]

We have added the following to section 3.1 line 210:

*"The ensemble median was nearly equivalent to the ensemble mean (r² = 0.99; not shown)."*

Line 210: "12.9 +:-1.1 µatm" : I'm not sure where this number comes from.

Added in text citation to the plotted data for this value to revised supplemental Figure S4, now including a second panel (b) with each ensemble member plotted against the consistent independent withheld data across all ensemble members.

[Figure]

*Figure S4 10-fold cross-evaluation (Section 2.4) individual ensemble member estimated $pCO_2$ against the (a) independent withheld data, and (b) 10% 10-fold evaluation data specific to that ensemble member. Mean root mean squared error (RMSE) and coefficient of determination ($r^2$) are across all individual ensemble members. Data is binned into 2 $\mu$atm by 2 $\mu$atm bins. The dashed black line represents a perfect fit of slope ($c_1$) = 1 and intercept = 0.*

Line 332: flux densities as high as 3.6, but your figures end at a maximum of 3?

Figure 7a groups data into seasonal (3 month) bins. The 3.6 value refers to just one month being January of 2000. Removed in text figure citation to avoid confusion.

Lines 394-395: provide some insight into how.

We have added example approaches to section 4.3 line 394 to help disentangle these distinct influences:

*"Unravelling the individual influence of these interconnected drivers (i.e., marine heatwaves, sub-decadal variability, and long-term trend) is not possible with this product alone but does prompt future inquiry in combination with regional models and emerging climate analysis tools (e.g., Chapman et al. 2022)."*

In this section you compare an atmospheric xCO2 with an increase in oceanic pCO2. Can't you convert the atmospheric xCO2 to pCO2 to compare things in the same ranges/units?

pCO2ATM = [PT - (RH/100) × PH2O] × xCO2ATM

where PH2O is the water vapour pressure at atmospheric temperature for xCO2ATM (in atm) calculated according to Dickson et al. (2007), RH is the relative humidity (in %) and PT is the total atmospheric pressure (in atm). If some of these are missing, you can get them from products (SeaFlux mentioned above). Otherwise you are comparing things that are not directly comparable.

Thanks for catching this error. It occurred in both in the legend and text. The values we refer to are atmospheric $pCO_2$ in µatm. The legend in figure 9 and text in section 4.4 have been corrected.

Line 437: detail how you got the vertical velocities with Ekman pumping

We have revised our text so it now includes in text citation in section 4.4 line 437 to MATLAB Climate Toolbox *ekman* function:

*"We find a strong spatial correlation between the trend in ΔpCO2 and the calculated average vertical velocity associated with Ekman pumping ($r^2$ = 0.64, p <0.01; Figure 11b). Ekman pumping was calculated from monthly, 1/4° spatial resolution Cross-Calibrated Multiplatform zonal and meridional ocean surface wind speeds (Mears et al. 2019) interpolated to 1/12°, using the MATLAB Climate Toolbox ekman function (Greene et al., 2017, 2019; Kessler et al., 2002)."*

Technical corrections

Abstract: For readers not familiar with your particular area, it would be better to introduce the fact that your area is a sink earlier.

Revised line 14 in the abstract to include this regional context:

*"Here we use a neural network approach to interpolate sparse observations, creating a monthly gridded seawater partial pressure of CO2 (pCO2) data product from January 1998 to December 2019, at 1/12°x1/12° spatial resolution, in the Northeast Pacific open ocean, considered a net sink region."*

Line 58: "The estimated long-term trend in surface ocean pCO2 appears to be increasing at less than the atmospheric rate." Less what? rate, amount of increase? Rewrite to clarify: "at a slower rate" or "less than the atmospheric rate of increase".

Revised to:

*"The estimated long-term trend in surface ocean pCO2 appears to be increasing at less than the atmospheric rate of increase (Franco et al., 2021)."*

Line 119: fCO2 has been converted to pCO2: how, give equation.

Revised to include in text citation to text added to the supplementary.

Supplementary Text TXX

*"The reported fCO2 estimates were converted to pCO2 using the equation SXX (Körtzinger, 1999):*

$$pCO_2 = fCO_2 \times exp[P_{atm}^{surf} \frac{B+(2\delta)}{RT}], \hspace{2cm} (SXX)$$

*where $P_{atm}^{surf}$ is the total atmospheric surface pressure, B and δ are viral coefficients (Weiss, 1974), R is the gas constant and T is the absolute temperature. National Centers for Environmental Prediction (NCEP) monthly mean sea level pressure was used for $P_{atm}^{surf}$ (Kalnay et al., 1996)."*

Figure 3: Add some metrics to the graphs (from the text)

Revised figure to include text with mean absolute error values for both products to compare more easily.

[Figure]

Line 150: "changing the internal division ratio", what did you end up using (see where you give more details).

Revised text to include in-text citation to later section.

*"… changing the internal data division ratio (94:6; see Section 3.4 below)."*

Line 424: "could be partially tied to if" missing word/rewording

Thanks for catching an error in the description. Revised text to:

*"Date combinations resulting in trends exceeding the atmospheric increase could be partly attributed to start and end dates coinciding with periods of weak and strong Alaskan Gyre upwelling, respectively. These upwelling modes induce negative and positive pCO2 anomalies, which further amplify the observed trend."*

Figure 7: the black line is not really black.

Revised figure so line is black.

Supplementary Figure 1 a&b: useful to superimpose bar charts of the whole dataset behind to further demonstrate how your subsampled data is representative.

Revised figure to superimpose bar charts of the training dataset behind.

---

## Author Comment (AC2)

**Duke et al. Reply on RC2: 'Comment on egusphere-2023-870', Anonymous Referee #2**

Summary:

Duke and coauthors use a two-step cluster–regression method to map surface partial pressure of carbon dioxide (pCO2) in the Northeast Pacific Ocean. Their approach is novel in that they grid pCO2 observations and produce pCO2 maps with high spatial resolution (1/12°); in doing so, they offer insightful observations about optimal model parameters and regional driving factors of CO2 flux variability. This work represents not only a useful product for investigating surface carbonate chemistry in the Northeast Pacific (NEP), but a valuable roadmap for increasing the spatial resolution of observation-based surface ocean pCO2 products.

This manuscript is very well-written and clear to follow. I was especially impressed with the analysis surrounding the training of artificial neural networks with progressively finer resolution, and the critical nature of the training/evaluation data split in these instances. The examination of driving factors of CO2 uptake variability and the effects of marine heatwaves is interesting and will be beneficial for researchers seeking a region-wide carbonate chemistry context for the NEP. I detail a few general and line-specific suggestions below, but overall support the acceptance of this manuscript.

Thank you for your time and careful consideration providing feedback on our manuscript. We appreciate your encouragement regarding the potential of our study to serve as a template for global products aiming to achieve higher spatial resolution. Below, we have addressed your comments in a point-by-point manner. Our responses are highlighted in blue, with manuscript text in quotations and added/revised text italicised.

General suggestions:

The conclusion that the training data to internal evaluation data ratio should be optimized and likely increased toward finer resolution grids will be extremely valuable as global-scale observation-based pCO2 data products with finer than 1° resolution are beginning to be produced. In that context, it may be helpful to expand upon the statement at the end of section 3.4 that this result "creates a precedent for stepping to a higher resolution product with nearly no loss in performance". How might you envision that higher resolution step being taken at a global scale? What are some important considerations and potential pitfalls when taking this approach beyond the NEP? Any thoughts about increases to the temporal resolution?

One important consideration with the NEP is that at 1/12° spatial resolution, gridded observation coverage is still actually quite good at 0.39%. In contrast, when looking at the SOCAT global coverage map (https://socat.info/), much of the south Pacific, south Atlantic, and Indian Ocean likely experience a more profound drop off in coverage compared to our Figure 4f "% target pixel coverage" line. Products would be relying on robust nonlinear relationships from the neural network informed in other regions to fill these gaps, raising a flag about critical observational coverage some have covered in the southern ocean (Hauck et al. 2023; https://doi.org/10.1098/rsta.2022.0063). Increasing temporal resolution is difficult and will likely need to be accompanied with changes to how we train the neural network. In increasing temporal resolution, the "% target pixel coverage" line in Figure 4f would move all values closer to zero. We feel the predictor training data may need to shift to using *in situ* data from high frequency underway systems prior to gridding to establish nonlinear relationships in the neural network,

before labeling higher temporal resolution gridded predictor data with estimated $p$CO$_2$ values. This approach could mark a major shift in the practice of creating observation-based products and significantly increase in computing costs.

Considering this discussion, the following text has been added to section 3.4 line 280.

*"In regions with sufficient observational coverage (Figure 4f; Bakker et al., 2016), this finding creates a precedent for stepping to a higher resolution product with nearly no loss in performance, overcoming the overfitting concern with increased resolution (Rosenthal, 2016)."*

One limitation of the validation performed here is that the statistical metrics represent the ability of the ANN-NEP procedure to estimate pCO2 only at the spatiotemporal grid cells where observations are available. This may mask location-specific seasonal biases, especially at high latitudes where wintertime observations are likely not as plentiful. In lieu of a comprehensive model simulation experiment to evaluate these unquantified biases, this consideration may warrant some discussion in section 3.2 or elsewhere.

RC2 is correct. We added the following text to section 3.2 line 225.

*"One limitation of our approach in assessing the uncertainty of the ANN interpolation method is that it is only applicable to grid cells where observations are available. Consequently, location-specific seasonal biases, especially in high latitudes with limited wintertime observations (Figure 1a&b), may not be fully captured or accounted for."*

A figure displaying the most frequent occurrence of each SOM province over the timeseries would be informative. As an additional suggestion for future work: to reduce the discontinuities at the borders of biogeochemical provinces it would be interesting to explore soft clustering approaches in addition to hard clustering like SOMs. Soft clustering approaches provide probabilities for each clustered grid cell, which can be used as weights to blend pCO2 predictions across clusters.

A figure displaying the mode SOM biogeochemical province in each pixel has been added to the supplementary and in text citation section 2.4 line 139.

[Figure]

*Figure S2 Mapped (a) mode of SOM biogeochemical provinces (i.e., most frequent occurrence), and (b) the number of unique SOM biogeochemical provinces each pixel belongs to for each month from January 1998 to December 2019.*

Moving to a soft clustering approach to remove artificial fronts created by the province boundaries is a great suggestion for future work.

Line-by-line comments:

Line 85: It would be valuable to articulate why the coastal ocean was excluded in this study.

The coastal ocean experiences much greater variability and presents all sorts of unique challenges. To create a "good" coastal product for the region, we felt that specific tuning measures would be needed. These issues will be addressed in a separate stand-alone paper led by PD – and is currently in progress. We have added to the text in section 2 line 85:

*"We limit our study region to the open ocean regions with reduced variability and related drivers compared to the continental shelf regions. Creating a product on the continental shelf and in the nearshore requires different neural network considerations and is associated with high uncertainties (Roobaert et al. 2023)."*

Lines 142–143: It isn't immediately clear why choosing not to normalize predictor data implicitly weights the SOM predictors toward the pCO2 climatology. Is it related to the relative range of each chosen predictor?

Yes, in not normalizing the SOM predictors data we forced the relative weights of the input data toward the pCO2 climatology, as the range between the lowest and highest value of pCO2 is at least one order of magnitude larger than that for SST, SSS, and log(MLD).

| Climatological SOM predictor variable | Mean ± standard deviation | Range |
|---|---|---|
| Climatological sea surface $p$CO$_2$ (µatm) | 355±18 | 209 |

| | | |
|---|---|---|
| Sea surface temperature (SST; °C) | 10±3 | 14 |
| Sea surface salinity (SSS) | 32.6±0.1 | 0.7 |
| Log mixed layer depth (log(MLD)) | 3±1 | 2 |

We added the following text to section 2.4 line 143.

*"We did not normalize predictor data (e.g., force a mean of 0 and standard deviation of 1), implicitly weighting SOM predictors toward the pCO2 climatology as its range is at east one order of magnitude greater than that of SST, SSS, and log(MLD) (Landschützer et al. 2013)."*

Lines 151–152: I don't understand what is meant by "we introduced each predictor variable again after deseasonalizing". Can this be explained more clearly?

In total we use 12 predictors in the FFN regression step being all those in table 1 plus those in table 1 deseasonalized:

1. Sea surface temperature (SST)
2. Chlorophyll-a (Chl)
3. Sea surface salinity (SSS)
4. Sea surface height (SSH)
5. Mixed layer depth (MLD)
6. Atmospheric pCO2
7. Sea surface temperature anomaly (SST)
8. Chlorophyll-a anomaly (Chl)
9. Sea surface salinity anomaly (SSS)
10. Sea surface height anomaly (SSH)
11. Mixed layer depth anomaly (MLD)
12. Atmospheric pCO2 anomaly

Update the text to include:

*"To emphasize interannual and longer-term trends within the six predictor variables (Table 1), each predictor variable is used in two different forms, first in its raw form and second after deseasonalizing, bringing the total number of FFN predictors used to 12."*

Lines 280–281: Very interesting and insightful conclusion!

Thank you.

Lines 455–456: It would be good just to clarify in this sentence that "stepping to a significantly higher spatial resolution" refers to a higher resolution "than typical observation-based pCO2 products (1/4° or 1° resolution)" or something along those lines.

Revised the text to include:

*"*We found that stepping to a significantly higher spatial resolution*, compared to typical open ocean observation-based pCO2 products (1/4° or 1° spatial resolution),* led to nearly no loss in performance despite a much lower ratio of gridded pCO2 observations compared to the total number of grid cells.*"*